# Evolution of protein-coupled RNA dynamics during hierarchical assembly of ribosomal complexes

Sanjaya C. Abeysirigunawardena[1,8], Hajin Kim [2,3], Jonathan Lai[4], Kaushik Ragunathan[5,9], Mollie C. Rappé[1,10], Zaida Luthey-Schulten[4], Taekjip Ha [1,5,6,7] & Sarah A. Woodson [1]

Assembly of 30S ribosomes involves the hierarchical addition of ribosomal proteins that progressively stabilize the folded 16S rRNA. Here, we use three-color single molecule FRET to show how combinations of ribosomal proteins uS4, uS17 and bS20 in the 16S 5' domain enable the recruitment of protein bS16, the next protein to join the complex. Analysis of real-time bS16 binding events shows that bS16 binds both native and non-native forms of the rRNA. The native rRNA conformation is increasingly favored after bS16 binds, explaining how bS16 drives later steps of 30S assembly. Chemical footprinting and molecular dynamics simulations show that each ribosomal protein switches the 16S conformation and dampens fluctuations at the interface between rRNA subdomains where bS16 binds. The results suggest that specific protein-induced changes in the rRNA dynamics underlie the hierarchy of 30S assembly and simplify the search for the native ribosome structure.

[1] T.C. Jenkins Department of Biophysics, Johns Hopkins University, 3400 N. Charles St., Baltimore, MD 21218, USA. [2] School of Life Sciences, Ulsan National Institute of Science and Technology, Ulsan 44919, Republic of Korea. [3] Center for Genomic Integrity, Institute for Basic Science, Ulsan 44919, Republic of Korea. [4] Department of Chemistry, University of Illinois at Urbana-Champaign, 600S. Mathews Avenue, Urbana, IL 61801, USA. [5] Department of Physics, Center for the Physics of Living Cells and Institute for Genomic Biology, University of Illinois at Urbana-Champaign, Urbana, IL 61801, USA. [6] Department of Biophysics and Biophysical Chemistry and Department of Biomedical Engineering, Johns Hopkins University, Baltimore, 21205 MD, USA. [7] Howard Hughes Medical Institute, Baltimore, MD 21205, USA. [8] Present address: Department of Chemistry and Biochemistry, Kent State University, Kent, OH 44242, USA. [9] Present address: Department of Biological Chemistry, University of Michigan Medical School, Ann Arbor, MI 48103, USA. [10] Present address: Sandia National Laboratory, Sandia, 87185-1468 NM, USA. Sanjaya C. Abeysirigunawardena and Hajin Kim contributed equally to this work. Correspondence and requests for materials should be addressed to T.H. (email: tjha@jhu.edu) or to S.A.W. (email: swoodson@jhu.edu)

During biosynthesis of the bacterial 30S ribosome, 21 unique ribosomal proteins bind the 16S rRNA in a hierarchy that ensures each rRNA assembles into a complete complex capable of normal protein synthesis[1, 2]. In the current model for assembly, each ribosomal protein stabilizes the native structure of one region of the 16S rRNA, enabling other proteins to join the complex[3]. For example, structural and biophysical studies showed that protein uS15 preferentially binds the folded conformation of a three-helix junction in the 16S central domain[4, 5]. uS15 binding also pre-organizes an adjacent helix junction[6], lowering the entropic penalty for binding the next proteins in the assembly map[7, 8]. Although such "progressive stabilization" models explain why protein binding stabilizes the rRNA in its native conformation, certain ribosomal proteins, such as uS4 and bS16, are indispensable for assembly[9, 10] even in $Mg^{2+}$ concentrations sufficient to fold the rRNA in the absence of protein[11]. Such ribosomal proteins must switch the rRNA into a different ensemble of structures that are capable of binding the next proteins.

To understand how ribosomal proteins fold the rRNA, we previously used smFRET to visualize encounters between ribosomal protein uS4 and the rRNA[12]. Protein uS4 (hereafter S4; tan surface in Fig. 1a) recognizes a five-helix junction (5WJ) in the 16S 5′ domain and is required to nucleate assembly of the 30S ribosome[13]. Our smFRET results showed that S4 and the 5′ domain RNA initially form randomly fluctuating encounter complexes that proceed through a "non-native" intermediate in which 16S helix 3 (h3; teal in Fig. 1) flips away from protein S4. After 1–2 s, the S4-rRNA complexes reach a slow dynamic equilibrium between the flipped intermediate complex and the native complex, in which h3 is docked against S4 as observed in the mature ribosome (Fig. 1a). Productive complexes access both conformations, and in this context, we use the term "native" simply to designate the conformation in the mature ribosome.

The 16S 5′ domain is the first region to be transcribed, and intermediate ribonucleoprotein (RNP) complexes containing the 5′ domain proteins appear early during 30S assembly[14, 15]. In addition to protein S4, proteins uS17 (S17) and bS20 (S20) each bind three- and four-helix junctions, whereas protein bS16 (S16) binds the interface between the S4 and S17-S20 subdomains (Fig. 1a). Although these four proteins do not contact each other in the ribosome, a web of rRNA tertiary interactions connects their binding sites so that addition of one protein is expected to influence binding of the next[16]. Assembly mapping experiments showed that protein S16 can only join the complex when S4 is present[17]. Because protein S16 is essential for later steps of 30S assembly[9, 18], a crucial question is how early binding proteins switch the complex from a state that is incompetent for assembly to one that can productively add the next proteins in the assembly hierarchy. Here, we use three-color single-molecule FRET[19] to directly observe the binding of multiple ribosomal proteins to the rRNA, and investigate the physical origins of cooperative assembly.

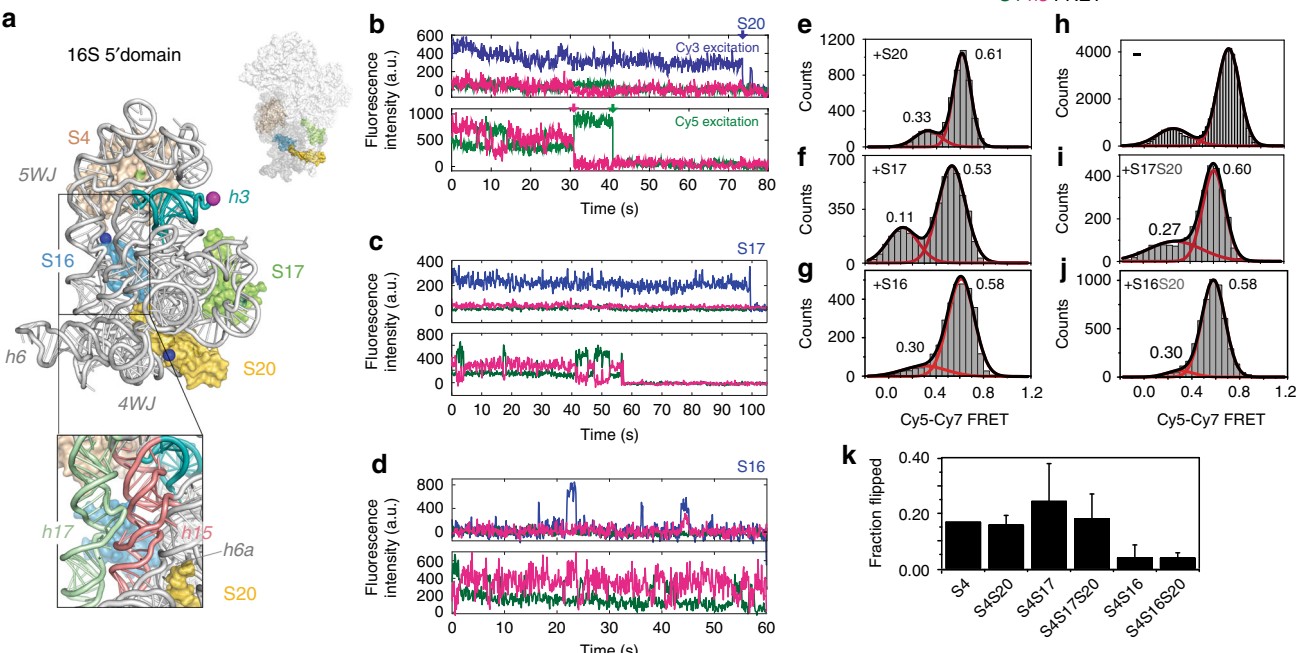

**Fig. 1** Ribosomal proteins change the preference for rRNA conformations. **a** *E. coli* 16S 5′ domain RNA (*gray ribbon,* main panel) forms the 30S body (small surface; PDB accession 2I2P[46]) and binds three primary assembly proteins (S4, S17, and S20) and secondary assembly protein S16. The RNA was fluorescently labeled with Cy7 (*magenta sphere*) by extension of helix 3 (h3; teal). S4 (*tan surface*) was labeled with Cy5 (*green sphere*). Proteins S16, S17, and S20 were labeled with Cy3 (*blue spheres*). RNA–protein complexes were excited by alternating 532 nm and 633 nm laser pulses using a custom-built multi-color single molecule FRET microscope[51]. *Inset*: expansion of S16 binding site showing h15 (*light red*) and h17 (*light green*). **b–d** Representative fluorescence traces obtained from complexes of 5′ domain RNA (h3-Cy7) and S4-Cy5 with S20-Cy3 **b**, S17-Cy3 **c**, or S16-Cy3 **d**. Cy3, *blue*; Cy5, *green*; Cy7, *magenta*. Single-step photobleaching events for each dye (*colored arrows*) indicate 1:1:1 stoichiometry between the components. S16-Cy3 exhibits high FRET efficiency to S4-Cy5 and h3-Cy7 upon specific binding to the complex (at 43 s in **d**). **e–j**. Histograms of FRET between S4-Cy5 and h3-Cy7 in the presence of the additional proteins in **e–g**, **i** and **j** were obtained from 110, 50, 30, 20, and 37 individual complexes, respectively. The Cy3 intensity was used to verify the presence of S20-Cy3, S16-Cy3 and S17-Cy3 **e–g**; the presence of unlabeled S20 **i**, **j** was inferred from the frequency of S20-Cy3 binding in **b**. Data in **h** are from Ref. [12] and represent Cy3-Cy5 FRET. **k** Population of the flipped conformation (low FRET) from the histograms in **e–j**. Error bars represent the s.d. between three data sets of each sample

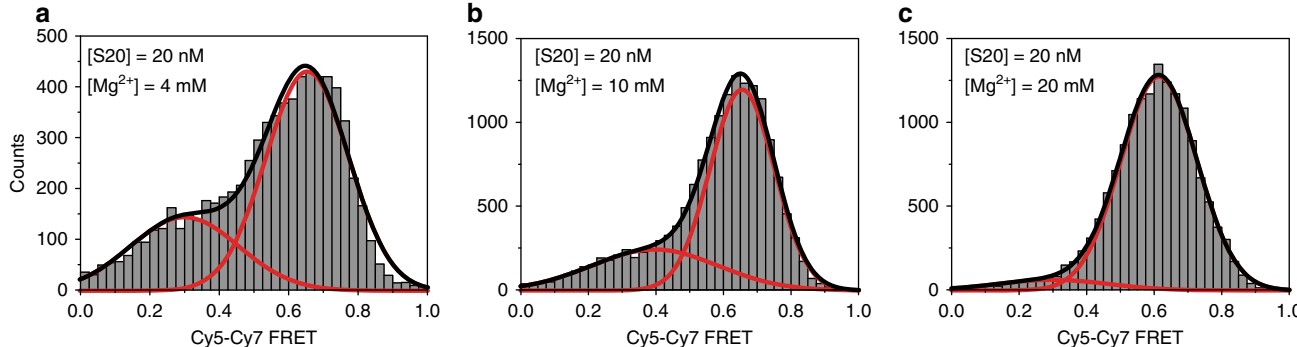

**Fig. 2** S4-S16 complexes sample the non-native conformation in physiological $Mg^{2+}$. Populations of native (high FRET) and flipped (low FRET) complexes containing 5′ domain h3-Cy7, S4-Cy5, and S16-Cy3 in 20 nM S20, at different $Mg^{2+}$ concentrations. Histograms for Cy5-Cy7 FRET are based on **a** 68, **b** 97, and **c** 105 trajectories from three-color smFRET experiments as in Fig. 1

## Results

**Conformational dynamics of multi-protein-rRNA complexes.** We first asked how each of the four ribosomal proteins that bind the 5′ domain RNA influence the exchange between native (docked) and non-native (flipped) conformations of 16S helix 3 (h3). Helix 3 connects the 5′ domain with the other domains of the 16S rRNA and must dock correctly for complete 30S assembly. We labeled h3 of the *E. coli* 16S 5′ domain by hybridizing a 3′ extension of the rRNA sequence to a DNA oligonucleotide modified with Cyanine7 (Cy7) fluorophore at its 3′ end, as previously described[12]. We attached Cyanine5 (Cy5) to protein S4, so that the docked form of h3 exhibits high FRET from S4-Cy5 to h3-Cy7, whereas the flipped intermediate exhibits low FRET between Cy5 and Cy7. In addition to S4-Cy5, the complexes contained proteins S16, S17 or S20 labeled with Cyanine3 (Cy3), or unlabeled S17 and S20 (Methods and Supplementary Fig. 1).

The various 5′ domain RNPs with three fluorescent labels were preassembled and immobilized on quartz microscope slides via a biotin on the 5′ end of the DNA oligonucleotide. The fluorescence intensity for each dye was recorded separately during alternating excitation of Cy3 and Cy5 as shown in Fig. 1b–d, which allowed us to measure all three pairwise distances between the three fluorophores[20]. We selected for analysis only those complexes that exhibited a single-step photobleaching event for each fluorescent dye (arrows, Fig. 1b), to ensure that they had the proper 1:1:1 stoichiometry. When S16-Cy3 bound the complex, we observed the expected energy transfer from S16-Cy3 to S4-Cy5 and h3-Cy7 in certain but not all cases (Fig. 1d). This energy transfer was used to select the trajectories in which S16 was bound to its specific site in the 5′ domain RNA. Because the binding sites for proteins S17 and S20 are 80 Å from the labeling sites on h3 and protein S4, too far to reliably observe energy transfer from S17-Cy3 or S20-Cy3 (Fig. 1b, c), we used co-localization of Cy3 with S4-Cy5 and h3-Cy7 to select complexes containing S17 or S20.

**Equilibrium between intermediate and native rRNA conformations.** In our three-color smFRET experiments, all of the 5′ domain complexes experienced transitions between the low FRET flipped conformation of 16S h3 and the docked state of h3, which resulted in high FRET from S4-Cy5 to h3-Cy7 (Fig. 1b–d). Therefore, h3 remains mobile even after proteins S4, S16, S17, and S20 have bound the rRNA. Histograms of the FRET populations for six combinations of the 5′ domain proteins showed that S16, S17, and S20 perturb the equilibrium between the docked and flipped conformations of h3, even though none of these proteins directly contacts h3 or S4. Although protein S20

had only a small effect on the equilibrium between flipped and docked h3 (Fig. 1e, h and Supplementary Fig. 2), addition of protein S17 increased the population of flipped intermediate (Fig. 1f, i), in agreement with ensemble FRET studies[21] and footprinting of 5′ domain complexes[16]. S17 binding also shifted the average FRET efficiency of the high FRET state from $E \sim 0.61$ to 0.53 and the low FRET population from $E \sim 0.33$ to 0.11 (Fig. 1f), suggesting S17 favors a different conformation of the 5′ domain RNA. By contrast, the low FRET peak shifted to $E \sim 0.27$ upon S20 binding. A similar upshift in the low FRET peak was observed upon binding of protein S16 (with or without S20), hinting that a change in the S20 region of the 5′ domain RNA is required for S16 binding. Protein S16 itself markedly stabilized the docked (native) conformation (Fig. 1g, j, k), consistent with our previous ensemble FRET results[21]. These conformational preferences illustrate how each ribosomal protein perturbs the energy landscape for rRNA folding.

These effects of the ribosomal proteins on the rRNA conformation were distinct from the stabilizing effects of $Mg^{2+}$ ions. $Mg^{2+}$ ions stabilize the docked high FRET state, and increase the kinetic barrier for exchange with the flipped low FRET state[12]. Binding of ribosomal proteins did not reduce this requirement for $Mg^{2+}$, because we observed transitions to the flipped intermediate state when the RNA was simultaneously complexed with S4, S16 and S20, especially in low $[Mg^{2+}]$ (Fig. 2 and Supplementary Fig. 3). Thus, unlike $Mg^{2+}$ ions, ribosomal proteins organize the rRNA structure, while still permitting exchange between alternative conformations.

**Binding kinetics of protein S16.** We next asked how the primary assembly proteins S4, S20, and S17 affect the ability of protein S16 to join the complex. In the Nomura assembly map, S16 binding requires the presence of S4 and is increased by the presence of protein S20. To observe S16 binding in real time, we tethered 5′ domain h3-Cy7 complexes with various combinations of proteins to the slide, and S16-Cy3 was injected into the slide chamber as the three-color fluorescence intensity was recorded. When S4 was omitted, we observed little or no Cy3-Cy7 co-localization, in agreement with the inability of S16 to bind the RNA in the absence of other proteins. When S16-Cy3 was added to RNA–S4 complexes, we observed co-localization of Cy3 with the immobilized complexes exhibiting FRET to S4-Cy5 and h3-Cy7, indicating site-specific S16 binding (Fig. 3a). However, these RNA–S4–S16 complexes were short-lived (~ 2 s), and many Cy3 complexes did not show the expected FRET levels, indicating that S16 often bound the RNA non-specifically. S16 binding in the presence of S4-Cy5 and 20 nM S17 was similarly short-lived (Fig. 3b).

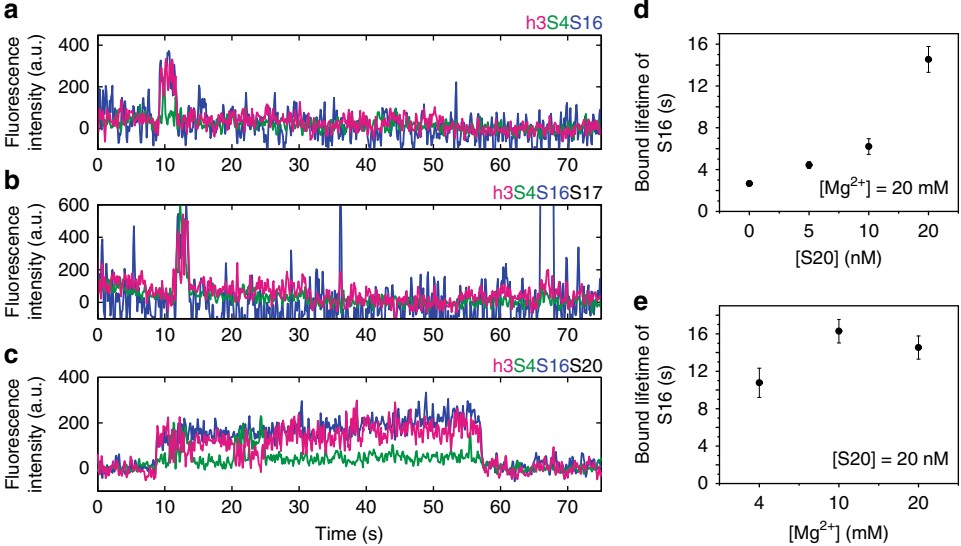

**Fig. 3** Lifetime of S16 binding depends on the composition of preassembled RNPs. **a–c** Examples of three-color fluorescence traces showing S16 binding in the presence of **a** S4, **b** S4 + S17 (20 nM), and **c** S4 + S20 (20 nM). These reveal the short lifetime of bound S16 when S20 is missing from the primary assembly complex. Fluorescence intensities are from S16-Cy3 (*blue*), S4-Cy5 (*green*), or h3-Cy7 (*magenta*) upon excitation of Cy3. The intensity jumps when S16-Cy3 localizes at immobilized preassembled 5′ domain RNPs. Anti-correlated fluctuations of Cy7 and Cy5 intensities indicate that 16S h3 remains dynamic after S16 binding. **d** Average lifetime of bound S16 vs. S20 concentration, in 20 mM MgCl$_2$. Each measurement is based on 75–154 molecules. Error bars indicate the s.e.m. from triplicate measurements. **e** Average lifetime of bound S16 vs. Mg$^{2+}$ concentration with 20 nM S20, as in **d**. Each measurement is based on 68–105 molecules. The lifetime of S16 binding strongly depends on the presence of S20 but not on the Mg$^{2+}$-dependent stability of the RNA structure

By contrast, S16 remained bound to the 5′ domain RNA significantly longer (≥10 s) in the presence of 20 nM S20 (Fig. 3c), demonstrating how S20 improves S16 recruitment to 30S complexes while S17 does not. The lifetime of RNA·S4·S16·S20 complexes in our experiments is likely sufficient for further 30S assembly, based on known rRNA folding and protein binding rates[22–24], and is consistent with the Nomura assembly map[17]. Raising the concentration of protein S20 from 0 to 20 nM extended the average lifetime of the S16 complexes about five-fold, from 3 s to 15 s at 20 mM [Mg$^{2+}$] (Fig. 3d). The stabilizing effect of S20 is likely greater than five-fold, because the longer S16 lifetimes are underestimated owing to fluorophore photobleaching. Mg$^{2+}$ ions, on the other hand, stabilize the rRNA tertiary structure[11] but had less impact on the lifetime of S16 binding than protein S20 (Fig. 3e). That Mg$^{2+}$ ions alone cannot recapitulate S20's effect suggests that S20 does not simply reinforce pre-existing native RNA tertiary interactions, but instead switches the 5′ domain RNA to a new structure that is competent for binding S16.

**Allosteric effect of S16 on the rRNA in real time**. Single molecule measurements can reveal the dynamic changes in biomolecules at the exact moment of encounter even when binding does not occur immediately after the components are mixed. We theorized that S16 might preferentially bind the complex when it is already in the high FRET state. Alternatively, S16 might bind either the low or high FRET state, but stabilize the high FRET only after the initial encounter. Using our three-color detection scheme, we exploited the FRET signal between S4-Cy5 and h3-Cy7 to observe the change in S4-h3 dynamics at the moment of S16-Cy3 binding. Figure 4a shows a representative time trace of S4-Cy5·h3-Cy7·S20 complexes as S16-Cy3 is added to the slide chamber. After the injection of S16-Cy3 solution (first arrow), several unsuccessful binding trials, represented by short spikes in Cy3 intensity, were observed before the productive binding event (second arrow).

The Cy5-Cy7 FRET trace upon Cy5 excitation revealed little change at the moment of the successful S16 encounter; S16 was able to bind to either the low or high FRET state, and h3 continued to fluctuate after S16 binding. However, when we synchronized a number of FRET traces at the S16 binding moment and overlaid them to build a time-dependent map of FRET population, the 2D histogram revealed a gradually decreasing population of the low FRET state after S16 binding (Fig. 4b). Overall, the low FRET population decreased to about half of the initial value at 5 s after the successful encounter (Fig. 4c). At more physiological 4 mM [Mg$^{2+}$], which makes the rRNA less stable and the low-FRET flipped intermediate state more visited, the low FRET population also decreased around the moment of S16 binding (Supplementary Fig. 4).

To quantify the effect of S16 on the h3 dynamics, we measured the change in the lifetime of the native (high FRET) and the intermediate (low FRET) states right before and after S16 binding. The lifetime of the high FRET state after S16 binding ($\tau_{post}$) was about 50% longer than before S16 binding ($\tau_{pre}$) (Fig. 4d, e). Conversely, S16 binding reduced the lifetime of the low FRET state by half (Fig. 4f, g), implying an increased rate of transition from the low FRET state to the high FRET state. Thus, S16 binding stabilizes the native rRNA structure, consistent with the shift in the FRET histogram (Fig. 1k). However, h3 continues to fluctuate between its native and non-native conformations while proteins S4, S16 and S20 are bound.

**Changes in RNA flexibility from simulations and footprinting**. Molecular dynamics (MD) simulations of the 5′ domain provided a structural explanation for this allosteric effect of protein S16 on h3 dynamics, which is communicated indirectly through 16S h12 (Fig. 5a). When bound, S16 contacts the base of h12, forcing the tip of h12 to pack against the minor groove of h3. These conserved and energetically favorable interactions between h12 and h3[25] prevent h3 from moving away from the 5WJ, and instead favor the native, high FRET state. In simulations without

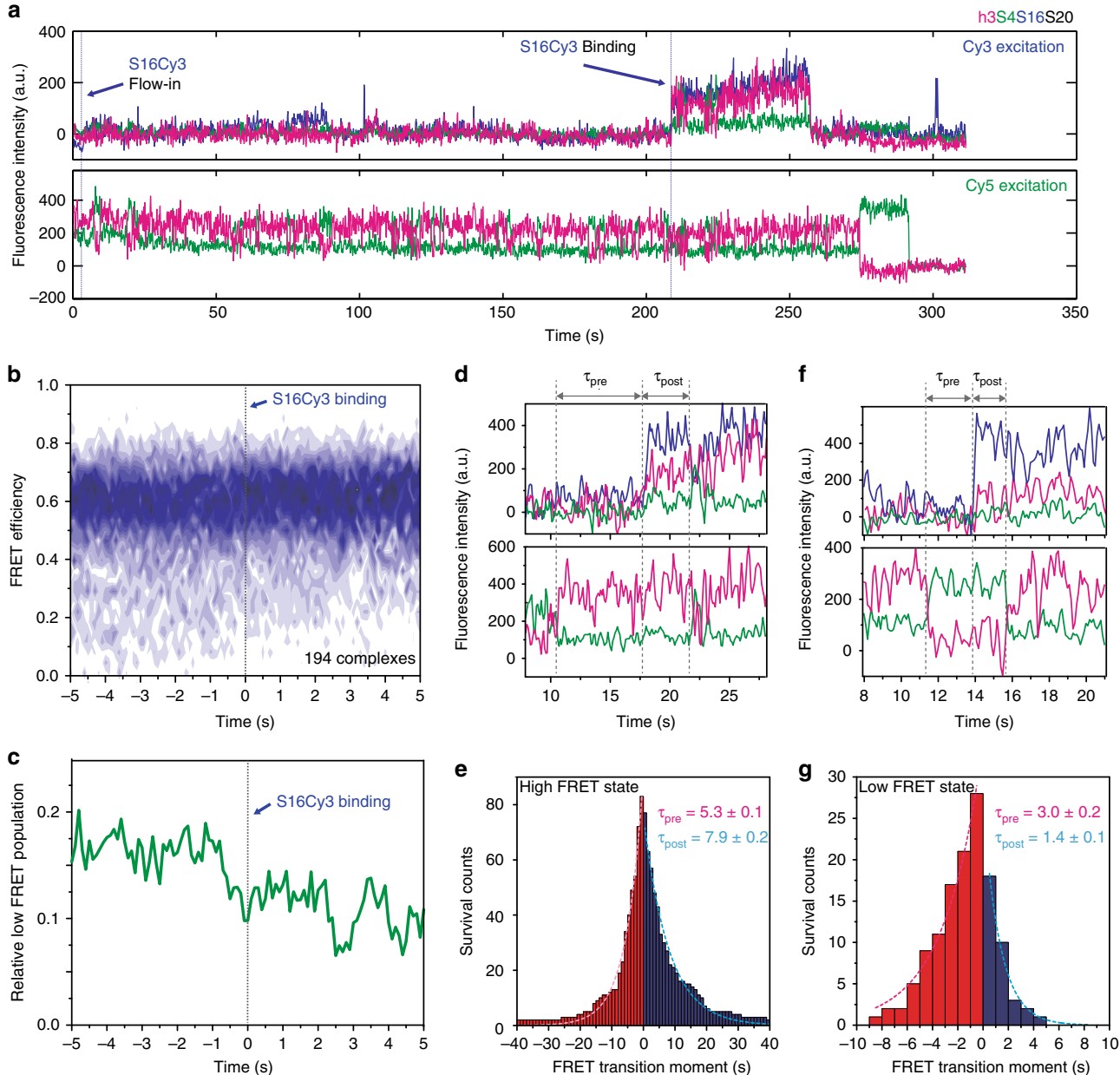

**Fig. 4** S16 indirectly induces the folding of the RNA. **a** Real-time binding of S16 to RNA–S4–S20 complexes. S16-Cy3 was flowed into slide chambers containing immobilized RNA h3-Cy7·S4-Cy5·S20 complexes. Three-color fluorescence traces with alternating Cy3 and Cy5 excitation demonstrate successful S16 binding following several trials, as indicated by the increase in Cy3 signal (top). 16S h3 fluctuates before and after S16 binding (bottom). **b** Time-dependent population map of S4-h3 FRET was obtained from 194 traces that are post-synchronized at the moment of S16 binding (dashed line). **c** Low FRET population ($E < 0.35$) from the map in **b** gradually decreases before and after S16 binding, demonstrating the preference of S16 for binding the complex in the high FRET state and its additional stabilization of the high-FRET complex. **d–g** Lifetimes of the high and low FRET states of h3 before and after the moment of S16 binding were compared. **d** The lifetimes of the high FRET state were calculated from the interval between S16 binding and the low-to-high FRET transition prior to it ($\tau_{pre}$) or the high-to-low FRET transition following it ($\tau_{post}$). **e** The average lifetime of the high FRET state slightly increased upon S16 binding (83 traces). **f, g** Analysis as in **d, e** showing that the lifetime of the low FRET state decreased significantly after S16 binding (28 traces)

S16 (S4 only or without proteins), h12 separates from h3 (toward the h12′ position in Fig. 5a) and allows h3 to move away from the 5WJ and S4, leading to the non-native low FRET state (Fig. 5b). Thus, our combined simulations and smFRET results suggest that S16 can bind the 5′ domain in both the flipped and docked h3 conformations. Once bound, S16 favors the native h3-h12 conformation by extending the lifetime of the h3 docked state and shortening the lifetime of the non-native flipped state.

To experimentally probe how S16 binding alters the flexibility of the 5′ domain RNA, we used SHAPE chemical modification of the ribose 2′ OH to compare the flexibility of the RNA backbone in the presence and absence of S16 (Supplementary Fig. 5). Nucleotides in 16S h7 and h11, which form part of the S17 binding domain, were more protected from SHAPE modification in reactions with proteins S4, S17, S20 and S16, relative to reactions with S4, S17, and S20 only (*blue*; Fig. 5c)[18]. This increased protection is consistent with more stable interaction

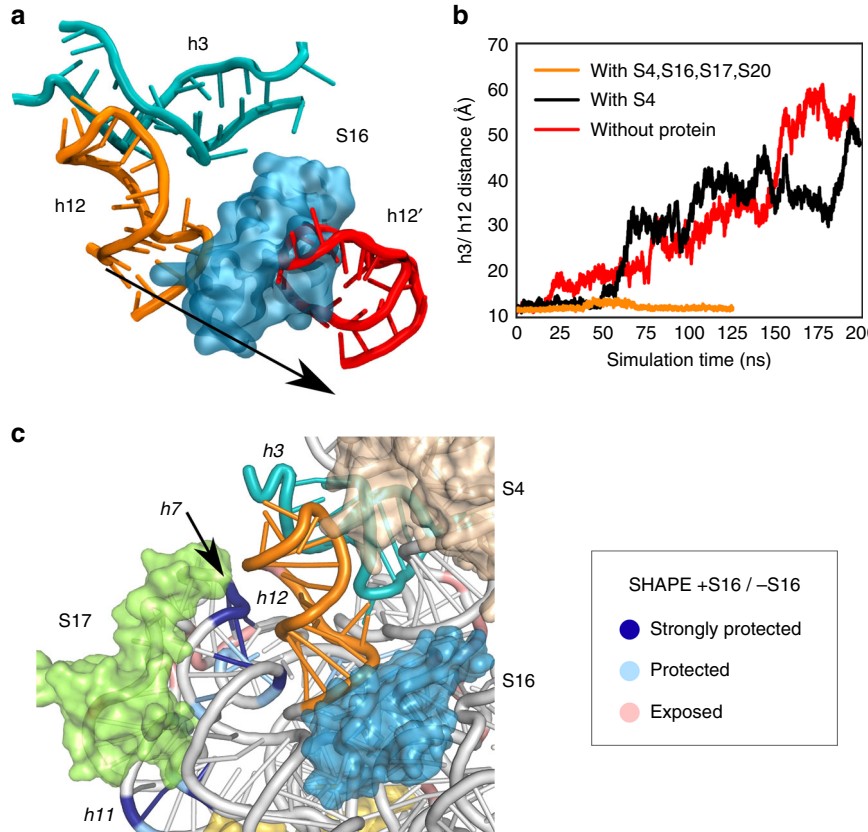

**Fig. 5** Molecular dynamics simulation of 5′ domain. **a** Snapshots from two molecular dynamics trajectories of h3 (*cyan*) and h12 (*red*: without protein; *orange*: with S4, S16, S17, S20). **b** Center of mass distance between residues at the interface of h3 and h12. When S16 is bound to the 5′ domain, h12 is pressed against h3 and the two helices remain packed together. Without S16, however, h12 becomes more pliable and dissociates from h3, causing the helix conformation to deviate from the crystallographic structure (Supplementary Fig. 5). **c** Differences in SHAPE modification caused by protein S16. SHAPE chemical modification of the ribose 2′OH increases with backbone flexibility and accessibility. Colors show the relative 5′ domain RNA modification $\rho$ in the presence of S4, S16, S17 and S20 (+S16) compared to S4, S17, and S20 (−S16): dark blue, strongly protected ($\log(\rho_{+S16}/\rho_{-S16}) = -1 \sim -1.5$); light blue, moderately protected ($-0.5 \sim -1$), pink, moderately exposed ($0.5 \sim 1$). See Supplementary Fig. 6 for further data

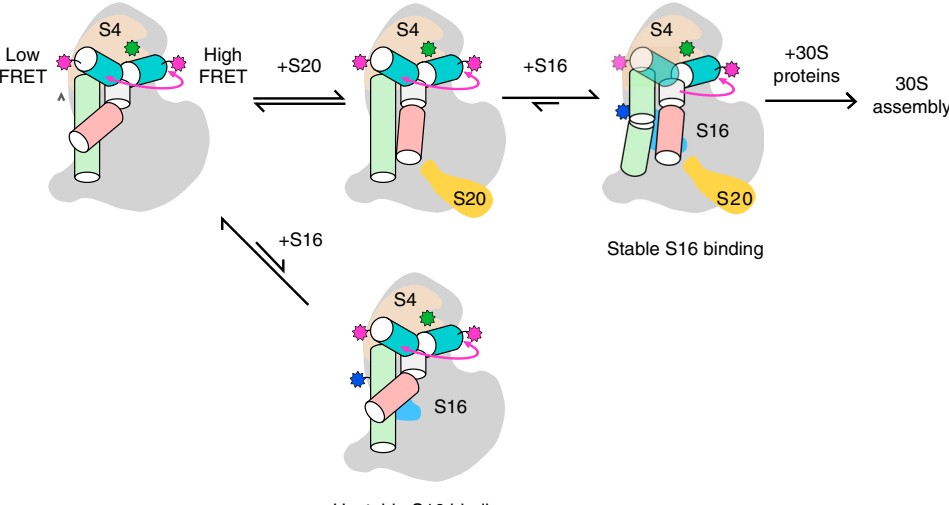

**Fig. 6** Protein-dependent switch in RNA dynamics promotes the hierarchical assembly of multi-protein complexes. Heterogeneously fluctuating encounter complexes between protein S4 (*tan*) and the 16S 5′ domain RNA (*gray*) transition to a slow equilibrium between a non-native low FRET conformation, in which 16S h3 flips away from S4, and a native high FRET conformation in which h3 docks against S4 (*pink arrow*). Binding of protein S20 does not alter the h3 equilibrium, but increases the probability of stable S16 binding by switching the conformation of h15 and neighboring helices (Supplementary Fig. 7). In the absence of S20, S16 dissociates from the RNA within a few seconds. Although h3 continues to fluctuate between low and high FRET conformations in S4–S20–S16 complexes, S16 binding progressively stabilizes the docked h3 complex that is competent to bind protein S12 during late steps of 30S assembly

between the S17 domain and the S4-5WJ domain when S16 is bound. Nucleotides at the 5WJ and in h15 are slightly more modified in the presence of S16 (red; Fig. 5c).

## Discussion

Many cellular complexes spontaneously self-assemble from their protein and nucleic acid components. Nevertheless, it was appreciated from early studies of protein folding that self-assembly cannot be achieved by a random conformational search, but must follow an energy landscape biased toward the native structure (reviewed in ref. [26]). The energy landscape for rRNA folding is shaped by the binding of ribosomal proteins that stabilize different regions of the rRNA. However, the real-time picture of assembly provided by the smFRET experiments shows that at least some RNA helices remain mobile after protein binding. Rather than rigidifying the RNA, each protein binding event alters the motions of the rRNA about specific helix junctions, changing the dynamics of the complex as well as its average structure. We suggest that fluctuations between different conformational states in the 16S 5′ domain, such as docking and undocking of h3, reorientation of h12, or a conformational switch in the S20 binding region, establish a preferred order of protein addition during 30S assembly.

In our three-color smFRET results, the probability of protein binding and the lifetimes of the complexes recapitulate the hierarchy of protein addition depicted in the Nomura 30S assembly map[17], in which protein S16 only forms specific complexes when S4 is also bound to the RNA. In our smFRET data, co-binding of S4 and S20, but not S4 and S17, increases the number of productive S16 binding events and the stability of S16 complexes (Fig. 6). Once bound, S16 shifts the docking equilibrium of 16S h3 toward the native conformation. Normal docking of h3 is crucial for 30S assembly, because h3 connects the 5′ domain to the central and 3′ domains in the 30S ribosome. Therefore, the ability of S16 to alter the h3 dynamics helps explain why S16 binding was discovered to be important for assembly of the 16S central domain (platform) in early footprinting experiments[18], and why S16 is needed for binding of protein S12 to h3 later in 30S assembly[9].

Although S20 and S17 both stabilize helix junctions in the 5′ domain RNA, only S20 transforms the RNA into a state that is competent to stably add S16. Molecular dynamic simulations and previous hydroxyl radical footprinting results provide evidence for an additional allosteric path between S20 and S16. First, molecular dynamics simulations of the 5′ domain RNA[27] showed that S17 and S20 have opposite effects on the RNA dynamics: S20 stabilizes the lower 4WJ and rigidifies the interface between the lower 4WJ and helices h15 and h17, which extend from the 5WJ bound to S4. By contrast, S17 increases the motions of the RNA helices at the subdomain interface where S16 must bind. Second, hydroxyl radical footprinting, which reports on the solvent accessibility of the RNA backbone, showed that the presence of S20 causes h6/6a (30S spur) to switch from a non-native to a native conformation as the 5′ domain RNA folds[16] (Supplementary Fig. 6). Because h6a packs against h15, this S20-dependent switch alters tertiary interactions in h15 and h17, also visible by footprinting, that are recognized by S16 (Fig. 1a inset). Transient repacking of h6/6a and h15 was not observed in the presence of S17, which favors different assembly intermediates than S20[16], consistent with our smFRET observation that S17 does not enable stable S16 binding.

The three-color smFRET results and MD simulations show that at least some ribosomal proteins act by redirecting the RNA motions between a narrower set of intermediates, rather than by locking rRNA helices in a fixed orientation. In support of this

model, we previously observed that rapidly fluctuating S4·RNA encounter complexes convert to a slow exchange between the flipped and native complexes after ~ 0.2 s (pink arrow, Fig. 6)[12]. Our three-color smFRET experiments now show that 16S h3 continues to fluctuate between the flipped and native conformations when S4 and S16, and likely S20 and S17, are bound to the same RNA molecule. S16 can bind either the flipped or the native S4·RNA·S20 complex (Fig. 4). Once S16 has bound, h12 becomes oriented toward h3, and transitions from the flipped to the native state become more probable (Fig. 4), giving rise to the observed shift in the population toward the native conformation (Fig. 1)[28, 29]. The shorter lifetime of the flipped state in the presence of S16 corroborates our earlier conclusion from ensemble binding experiments that S16 binding raises the free energy of the non-native complex more than it lowers the free energy of the native complex[28, 29].

These protein-induced changes in the RNA conformation and dynamics are consistent with ensemble models for allosteric interactions[30, 31]. Nevertheless, the smFRET experiments show that the ribosomal proteins alter the conformations sampled by a single RNA over several seconds, as the initial encounter complexes progress to the final complex. This progressive change in the RNA dynamics after the initial binding event, which is even more pronounced for S4[12], combined with the MD simulations and footprinting results, suggests that the system experiences a series of barrier crossing events that switch the RNA–protein complex from an assembly incompetent structure to one that is able to productively bind the next protein in the ribosome assembly hierarchy.

Although proteins S17 and S20 alter the likelihood of stably recruiting protein S16, neither protein is essential in E. coli, demonstrating that the rRNA itself folds well enough to pass by this step of assembly[32, 33]. This raises the question of what advantage these proteins offer that cannot be achieved by more stable RNA interactions. $Mg^{2+}$ ions, which often lower RNA folding rates[34, 35], slow the rate of exchange between the flipped and native h3 conformations[12]. By contrast, proteins S4 and S16 preserve the mobility of certain RNA helices, while selecting against unproductive conformations. We speculate that protein-guided switching of the RNA dynamics smooths the search for the native structure by selectively allowing the system to cross certain free energy barriers but not others. Similar protein-dependent dynamics may occur during the assembly of other RNA–protein complexes[4, 36, 37].

## Methods

**Ribosomal protein modification and labeling.** E. coli ribosomal proteins were over-expressed from pET24b derivatives in BL21(DE3) cells and purified by cation exchange chromatography (UNO S6, BioRad)[38] with a few modifications[29]. Single cysteines for labeling were introduced by site-directed mutagenesis (Quikchange) at positions that are not conserved (S16:S44C) or that have been shown not to interfere with 30S assembly (S20:S23C[39]). Protein S4:C32S,S189C was prepared by Quikchange mutagenesis previously[29]. Purified proteins were dialyzed overnight into 80 mM K-Hepes pH 7.6, 1 M KCl, 1 mM TCEP with three buffer changes and stored at −80 °C in 500 μL aliquots.

The mutant proteins were fluorescently labeled with a six-fold excess of maleimide-Cy5 (GE Healthcare) (S4 C189) or maleimide-Cy3 (S16 C44, S17 C53 and S20 C23) in 80 mM K-Hepes pH 7.6, 1 M KCl, 1 mM TCEP, 3 M urea at 20 °C[40]. Excess unreacted dye was removed by cation exchange chromatography and dialysis (three times) against 80 mM K-Hepes pH 7.6, 1 M KCl, 6 mM 2-mercaptoethanol. Protein concentrations were determined by absorption at 280 nm (unlabeled; $\varepsilon_{280,S4} = 17,843$ $M^{-1} cm^{-1}$, $\varepsilon_{280,S16} = 6990$ $M^{-1} cm^{-1}$, $\varepsilon_{280,S17} = 6990$ $M^{-1} cm^{-1}$, $\varepsilon_{280,S20} = 1490$ $M^{-1} cm^{-1}$), 550 nm (Cy3; $\varepsilon_{550} = 1.5 \times 10^5$ $M^{-1} cm^{-1}$) and 650 nm (Cy5; $\varepsilon_{650} = 2.5 \times 10^5$ $M^{-1} cm^{-1}$) respectively.

**Single molecule FRET measurement and analysis.** For three-color detection of ribosomal protein–RNA complexes, protein S4 was labeled with Cy5 at residue 189. A 3′ extended form of the E. coli 16S 5′ domain (nt. 21–556) was hybridized with a 68 nt oligonucleotide attached to Cy7 on its 5′ end and biotin on its 3′ end, and with a DNA complementary to the 3′ end of the 68-mer[12]. This arrangement

placed Cy7 adjacent to the 3′ end of 16S h3, and biotin ~ 60 Å away from 16S h3. The labeled S4-5′ domain RNA complexes were assembled in 30S reconstitution buffer (80 mM K-HEPES pH 7.6, 330 mM KCl, 4–20 mM MgCl₂, 6 mM 2-mercaptoethanol) and immobilized on quartz slides coated with PEG and neutravidin[41] through the biotinylated oligonucleotide[12]. S4 remains bound during our experiments ($\tau_{off} \geq 30$ min). Cy3-labeled S16, S17, and S20 were used at 20 nM (2–3 times $K_D$) unless stated otherwise. Three-color smFRET measurements were performed with alternating excitation by 532 nm and 633 nm lasers of 50 ms duration each[12]. The FRET efficiencies between three fluorophores were calculated after background subtraction, and after correcting for leakage and for the less efficient detection of Cy7. The corrections needed to calculate three-color FRET efficiencies are detailed elsewhere[20]. FRET histograms were obtained by inspecting each three-color trace and selecting traces that show 1:1:1 stoichiometry of all three labels, as judged from single-step unbinding or photobleaching, and frame ranges within these traces in which all three fluorescence signals are clearly detected. FRET histograms were fit (least squares) with double Gaussian distributions to obtain the population fractions. The lifetimes of protein bound states were measured by averaging over all detected events. Two-dimensional maps of the FRET histogram over time were constructed by synchronizing three-color traces exhibiting S16 binding events to the moment of S16 binding. This was defined as the moment the total fluorescence intensity rises beyond a threshold, after the traces were smoothed as a three-frame moving average. The threshold for all traces was set at the middle between the average total intensities before and after S16 binding.

**Molecular dynamics**. MD simulations (5′ domain with S4 only, S4, S16, S17, S20, and without proteins) were taken from a previously published work[27] and reanalyzed using VMD[42]. Proteins and nucleic acids were parameterized with the CHARMM22[43] with CMAP corrections and CHARMM27[44] force fields, respectively, using the minimization and equilibration protocol established in Eargle et al.[45]. Briefly, the system was prepared from the crystal structure 2I2P[46] and ionized using Mg²⁺ and K⁺. In total, all systems had 370,000 atoms. MD simulations were performed using NAMD 2.9[47]. Each system was run at 300 K, 1 atm for at least 100 ns using a 1 fs timestep and a 12 Å cutoff. Periodic boundary conditions and PME were used to evaluate nonbonded interactions. To monitor the interaction between h12 and h3, we calculated the center of mass distance between residues (G299, G301, A303) and (U555, G557, A559) in h12 and h3, respectively. These residues were selected based on network analysis, of which the details are available elsewhere[27]. Briefly, each MD system is coarse-grained into a network of nodes. Each node describes either the center of mass of an amino acid, a nucleobase, or a nucleotide sugar. Edges are defined for nodes that have a cutoff of < 4.5 A and have a trajectory occupancy of > 75%. Generalized correlations are calculated between each pair of nodes. These generalized correlations are converted into a distance metric ($d = -||\log(C_{ij})||$). We also identified the geometrically most central nodes in the proteins S4, S16, and h3 and determined the shortest path connecting these central nodes. These pathways passed through the aforementioned residues (G299, G301, A303) and (U555, G557, A559). When comparing networks from simulation with S4, S16, S17, S20, and without any proteins, the edges connecting these two sets of residues underwent the largest change in correlation, suggesting that these edges were important to stabilize the h3 and h12 interaction.

**SHAPE footprinting**. An extended form of the 16S 5′ domain RNA suitable for reverse transcriptase primer extension[48] was modified with 30 mM N-methylisatoic anhydride (NMIA)[49] 45 min at 37 °C. Before SHAPE modification, 4 pmol 5′ domain RNA was allowed to fold 15 min at 37 °C in HKM20 Buffer (80 mM K-Hepes pH 7.5, 330 mM KCl, 20 mM MgCl₂). To the RNA was added 8 µl Binding Buffer (80 mM K-Hepes pH 7.5, 330 mM KCl, 20 mM MgCl₂, 0.01% Nikkol, 6 mM β mercaptoethanol), or 8 µL Binding Buffer containing 16 pmol S4, 40 pmol S17, and 20 pmol S20, or 16 pmol S4, 40 pmol S17, 20 pmol S20, and 20 pmol S16. The RNA–protein mixture was incubated 45 min at 37 °C. These 1:4:10:5:5 ratios were empirically determined to saturate the protein–RNA interactions by test SHAPE titrations. The modified RNA was extracted with phenol and chloroform, precipitated with ethanol, and analyzed by extension of Beckman D4-labeled primers with SuperScript III Reverse Transcriptase (Invitrogen). The cDNA was analyzed on a Beckman CEQ-8000 with a D3-labeled library from unmodified RNA, an IR800-labeled ddCTP sequencing ladder, and a D2-labeled ddGTP sequencing ladder. The raw CEQ traces were processed with ShapeFinder to determine peak areas for each 5′ domain nucleotide[50]. After background subtraction, the peak areas were scaled such that the average of the 92–97 percentile peak areas equaled 100. All nucleotides with a reactivity of < 2.5 were set to a baseline value of 2.5. The log ratios of the reactivity, $\rho$, in the presence of protein S16 and without S16, were plotted on a histogram and clustered.

**Data availability**. The data that support the findings of this study are available from the corresponding author upon reasonable request.

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

## Acknowledgements

We would like to thank G. Culver for the gift of plasmids. This work was supported by the National Institutes of Health (R01 GM68019 to S.A.W. and R35 GM122569 to T.H.), the Howard Hughes Medical Institute (T.H.), and the National Science Foundation (MCB 12-44570 to Z.L.-S. and PHY 1430124 to Z.L.-S. and T.H.). H.K. was supported by the National Research Foundation of Korea (2014R1A1A1003949), IBS-R022-D1, and UNIST research fund (1.170009.01). J.L. was supported by the US Department of Energy, Office of Science, Biological and Environmental Research as part of the Adaptive Biosystems Imaging Scientific Focus Area.

## Author contributions

S.C.A. designed, prepared and validated samples, S.C.A. and H.K. acquired and analyzed smFRET data, K.R. contributed to data acquisition and instrument design, J.L. and Z.L.-S. performed MD simulations, M.C.R. performed SHAPE footprinting, S.C.A., H.K., T.H. and S.A.W. designed the experiments and wrote the paper.

## Additional information

**Competing interests:** The authors declare no competing financial interests.

