## [Peer Review File · Nature Communications]

Reviewers' Comments:

Reviewer #1 (Remarks to the Author):

The role of RNA conformational changes during assembly of a region of 16S rRNA is examined here by several complementary methods. Addition of proteins in the S4/S16/S17/S20 cluster is reported to result in a cascade of RNA rearrangements that facilitate progressive assembly. While some RNA conformational changes are correlated to direct binding of a proximal protein, others appear to be triggered by binding of a remote protein.

The three-color FRET experiments show unambiguously that the rRNA is sampling orientations as proteins are bound. The authors conclude that those reorientations are required for the ordered assembly of the RNP, and describe the assembly process as allosteric. There will be some people who argue about such demonstrations of allostery, since there are no thermodynamic measurements to quantify it. Including a short note about allosteric interactions seems appropriate. A reference to Hilser's work might help.

Many observations are reported that could be expanded upon. For example, the S20 vs Mg²⁺ contributions to h3 dynamics and S16 binding in Figure 3. These experiments were done at (presumably) saturating concentrations of S20 and Mg²⁺, but under conditions of lower concentrations, would there be inter-dependence (Figure 2 suggests the two ligands are not independent)? In Figure 4, is it correct to say that there are fewer RNA:S4 contacts when S16 is bound? If so, is that attributable to a change in the relative orientation of RNA and S4 or a rigidification of RNA so that it can't sample conformations?

In the context of the short manuscript, such expansions are not feasible. The authors might consider highlighting their novel observations.

The MD runs are very short (200 ns), and as Figure 5b shows, they are not converged; the distance between h3 and h12 appears to be increasing. h12' looks very distorted, as do all the helices. My confidence in conclusions based on these experiments is low.

In Figure 6, the authors present a model of the process, but the hydroxyl radical footprinting data report on a region of the complex that is only peripheral to the focus on h3 and S4. Are comparable data available for the regions highlighted in the model? In the text, make an explicit connection between the footprinting and the model shown.

There are frequent discrepancies between figures and figure legends that frustrated this reader. For example, Figure 5: 5a legend ...with (green) and without (blue) S16 - there is no green. 5c med and dark blue sites cannot not be distinguished (at least in my printout and the screen). In 5c, label h3. In Figure 6, there is no 6c. Figure 1a has no black or green spheres. Figure 4a colors aren't coordinated with text.

Reviewer #2 (Remarks to the Author):

Abeyirigunawardena et al. characterize the dynamics of the 5' domain of the small ribosomal subunit and four of its constituent proteins. Three color FRET is used to monitor multiple sites simultaneously in an attempt to match dynamical changes with protein binding. Molecular dynamics simulations and SHAPE experiments are used to correlate the coarse grained FRET information with structural changes. The global thermodynamic picture of the order in which the proteins bind to the small subunit is known as the Nomura map. The FRET experiments here are consistent with this picture and are able to add some kinetic details.

Overall, I think the data in this paper is valuable for the construction of new experiments and for comparison to simulations. Also, I agree with the authors that there are protein-coupled changes in dynamics and that these changes are strongly-related to the hierarchy detailed in the Nomura map. But I don't think that the data in this paper directly supports the claim of the title - "Assembly hierarchy of ribosomal complexes depends on protein-coupled RNA dynamics." The data characterizes the dynamics of the 16S in various protein bound stages, but is not able to isolate changes that are responsible for further protein binding (S16 binding in the case of this paper).

1. I think the three color FRET system developed here is well characterized and gives clean data. But, I don't think that the FRET data supports that protein-coupled changes in RNA dynamics are important for the binding hierarchy. The data related to the title is that S16 can bind more stably when S20 is already bound, which they indeed see with the FRET, but the FRET does not tell them how S20 changes the dynamics of the small subunit to allow S16 to bind. Fig 1 shows a FRET distribution +S16 vs +S16S20 which are the same. They show that S16 can (at least at higher than physiological Mg) change the dynamics of helix 3, but this is after S16 is already bound.

2. The SHAPE analysis also seems not to target the differences in small subunit conformation allowing S16 binding because they do not compare SHAPE with S20 and without S20 in the absence of S16. They show that there are global differences with and without proteins, but do not

try to isolate SHAPE differences between 16S assemblies that are favorable/unfavorable to S16 binding.

3. There is some discussion of later binding of S12 being affected by the dynamical changes imposed by S16 binding. These arguments make sense, but for this to be used in support of the central argument it must be explained what the evidence there is that S12 depends on these previous S16 binding changes. As far as I know, there are multiple events, S16 binding and the existence of the central domain before S12 can bind, which makes it difficult to identify the differences caused by S16 in isolation with S12 binding.

A couple technical questions:

4. In Fig 4 there is a typo "excitstion" in panel a. I am a bit confused by the colors of the Cy3/Cy5 excitation labels. Cy3 seems to be colored blue, but the Cy3 excitation label is written in green. Also the colors of the h4/S4/S16 don't seem to be consistent with S4 being bound to Cy5.

5. Why are the shifts in h3 dynamics upon S16 binding presented as histogrammed kinetic traces? If it is the case that S16 binding changes the nature of the small subunit, then the kinetics and thermodynamic shift in h3 FRET signal should be preserved for the entire dwell time of S16 bound to the small subunit. If so, then limiting your data set to just the binding events greatly limits the available data.

Reviewer #3 (Remarks to the Author):

Pioneering work of Nomura and Nierhaus in the 1970's showed that reconstitution of bacterial ribosomal subunits from their constituent RNAs and proteins was a hierarchical process. The stable association of some proteins with rRNA depended on the prior association of others. Subsequent work by them and many others has unravelled many details of rRNA folding dynamics, allosteric effects within the rRNP, and transitions of r protein-rRNA interactions from less stable "encounter complexes" to the final stable structure, etc.. However, we still do not completely understand the details of the fundamental mechanisms by which multiple proteins assemble with a long RNA molecule. For example, what underlies the hierarchy if the proteins do not directly contact each other? Here, Sarah Woodson and her coworkers use three color single molecule FRET, molecular dynamics simulations, and SHAPE assays to uncover mechanisms of the interdependency among bacterial ribosomal proteins S4, S17, S20 and S16 and rRNA to assemble with the 5' domain of 16S rRNA.

The take-home messages are impactful to the field of ribosome assembly, and are of broader interest for those studying RNA-protein interactions. The bottom line of this work is summarized best by the authors: that binding of an r protein does not simply stabilize pre-existing native tertiary structures, but alters the RNA dynamics to "make the search for native structure more efficient"..... "The experiments directly capture the dynamic nature of the allosteric interactions that underlie the cooperativity of ribosome assembly."

The manuscript is well written and generally accessible, even to those not aficionados of SM studies. The hypotheses, experimental rationale, and conclusions are elaborated clearly, in a stepwise fashion.

However, surprisingly, and in stark contrast, the Figure legends contain many trivial mistakes. Perhaps lost in transition to the last draft of the ms... I suggest that the authors spend some time carefully editing the legends. Some examples: Figure 1a: do the colors match those listed in the legend? Where is figure 1k? Also, since it is so critical to help readers understand this work, I suggest enlarging Fig.1a, showing the structure of the RNP domain being assembled. Fig.5: colors are not labeled correctly. Fig.6: there is no fig. 6c as indicated by the legend. Please also enlarge or highlight more the red arrow.

Reviewer #4 (Remarks to the Author):

In this manuscript, the authors examine the hierarchical interplay with which the ribosomal proteins S16, S4, S17, and S20 stabilize the folded 16S rRNA and assemble into the 30S ribosome, facilitating the formation of the native 30S ribosome structure. The authors utilize a combination of elegant 3-color single-molecule FRET and colocalization imaging approaches, footprinting and molecular dynamics simulations.

This manuscript represents a substantial and timely advance and reports exciting new data that fill major gaps in our understanding of those processes that enable rapid and efficient assembly of native ribosomes. Importantly, the data presented convincingly support the conclusions. In particular the single-molecule FRET experiments by the Ha laboratory are technically of high quality and well presented. This is a very challenging system to study in the context of a high-quality single-molecule experiment, and even more so since it involves 3-color FRET imaging.

However, the figures appear to be assembled in a haste and readability suffers. For example, the

labeling scheme in Figure 1a is not depicted in a clear manner. Are the green spheres that are referred to in the legend of Figure 1a green or blue? Where is the sphere for Cy3 on S17 in Figure 1a? The black sphere referred to in the legend appears purple in the structure. The legend for ‘b-d’ needs to be fixed. Similarly, the figure legend for Figure 3 states that ‘a-c’ show fluorescence traces, while this is clearly not the case, etc.

There are a large number of such inaccuracies throughout the manuscript, and a carefully revised version would greatly help the reader to appreciate the authors’ important experimental findings.

Aside from these minor issues, I believe that this will be a very important paper in helping our understanding of the hierarchical ribosome assembly move forward. As such I am happy to recommend publication of this manuscript in Nature Communications.

Reviewer #1 (Remarks to the Author):

The three-color FRET experiments show unambiguously that the rRNA is sampling orientations as proteins are bound. The authors conclude that those reorientations are required for the ordered assembly of the RNP, and describe the assembly process as allosteric. There will be some people who argue about such demonstrations of allostery, since there are no thermodynamic measurements to quantify it. Including a short note about allosteric interactions seems appropriate. A reference to Hilser's work might help.

Response: This is an important comment. We revised the end of the discussion on pg. 9 and added references to reviews by Csermly & Nussinov and Hilser on ensemble (and dynamical) models of allostery in proteins. We also cite our previous bulk measurements of S4 binding affinity in the presence of S16 and other proteins, which provides a thermodynamic view of these interactions.

An important point of this paper is that a classical thermodynamic framework is inadequate to describe our observation that the proteins alter the dynamic sampling of RNA conformations over time, as the initial encounter complexes progress to the final complex(es). Instead, it appears that the system experiences a series of barrier crossing events that move the RNA from one set of minima in the free energy landscape to another. It is not just that protein binding selects out the most stably folded RNA conformations – it also makes previously low-energy conformations unfavorable. This feature of the system was unexpected, but we believe it is a key mechanism by which ribosomal proteins smooth the assembly path to native ribosomes. We thank the reviewer for asking us to explain this critical point more clearly.

Many observations are reported that could be expanded upon. For example, the S20 vs Mg²⁺ contributions to h3 dynamics and S16 binding in Figure 3. These experiments were done at (presumably) saturating concentrations of S20 and Mg²⁺, but under conditions of lower concentrations, would there be inter-dependence (Figure 2 suggests the two ligands are not independent)?

Response: We have not covered all combinations of S20 and Mg²⁺ concentrations, so we cannot address this point in detail. There may be some interdependence between S20 and Mg²⁺ ions. Nevertheless, Figure 3d shows that even at the saturating Mg²⁺ concentration that should maximally stabilize the RNA structure, S16 binding strongly depends on protein S20. Conversely, high S20 concentration guarantees stable S16 binding regardless of Mg²⁺ concentration. Thus, our results indicate that a specific conformational change caused by S20, but not high Mg²⁺, is necessary and sufficient for stable S16 binding.

In Figure 4, is it correct to say that there are fewer RNA:S4 contacts when S16 is bound? If so, is that attributable to a change in the relative orientation of RNA and S4 or a rigidification of RNA so that it can't sample conformations?

Response: Our previous footprinting data suggest that S4 makes MORE contacts with helix 18 and h12 in the presence of protein S16, rather than fewer contacts. Figure 4 describes experiments in which S16-Cy3 is added to the slide chamber containing immobilized RNA-S4-S20-S17 complexes. These data show that the 16S rRNA remains dynamic throughout the assembly progress, sampling an ensemble of conformations. S16 binding does not necessarily prevent such dynamics and rigidify the RNA, but it rather shifts the balance toward the conformations with folded h3, which has an important implication for the downstream assembly process.

In the context of the short manuscript, such expansions are not feasible. The authors might consider highlighting their novel observations.

Response: We have revised the discussion section to better summarize the novel observations. The MD runs are very short (200 ns), and as Figure 5b shows, they are not converged; the distance between h3 and h12 appears to be increasing.

We agree with the reviewer that the MD runs are short and, in the simulations without S16, that the distance between h3 and h12 continues to increase over time and have not converged. The helices have separated beyond both the potential cutoff distance and the radius of gyration for the crystal structure (43 Å); h3 and h12 no longer interact with each other and are, effectively, freely diffusing from each other. If one assumes that the RNA of 5' domain is a 3D randomly-structured polymer, the theoretical expected distance between h3 and h12 is 56 Å---which is the value seen at the end of the MD simulation for the simulation without protein.

$$\begin{aligned} \langle r \rangle &\approx d_{base\ stacking} \sqrt{N} \\ &\approx 3.5 \text{Å} \sqrt{300} \\ &\approx 56 \text{Å} \end{aligned}$$

Thus, the results will probably converge if we extend the calculations out another few hundred nanoseconds.

Nonetheless, we do not expect that running the simulations out to convergence will change the conclusion of the MD section---namely that the h3 and h12 interaction is stable only in the presence of S16.

h12' looks very distorted, as do all the helices. My confidence in conclusions based on these experiments is low.

We agree with the reviewer that the helices appear distorted because the 5' domain without any proteins is "unfolding". Without any proteins, h3/h12 base pairing have started to switch from Watson-Crick (WC) to other types of base pairs (table below; as detected by x3DNA-dssr), and the root-mean-square-deviation (RMSD) of h3/h12 steadily increases over time (see Figure). It should be noted that h3 is the terminal helix in our molecular model and will be more inherently more flexible than h12.

h3	Num. of base pairs	Num WC:	Num. other
Crystal structure	11	9	2
Without protein	9	3	6
With S4, S17, S20, and S16	8	7	1
h12			
Crystal structure	10	6	4
Without protein	9	2	7
With S4, S17, S20, and S16	11	6	5

Upon adding the proteins S4, S16, S17, and S20, the 5' domain base pairing goes up and the RMSD of h3/h12 stays level at 2-3 Å. We added Figure S5 (see below) comparing the conformations of helices 3 and 12 with and without proteins to illustrate this point.

In Figure 6, the authors present a model of the process, but the hydroxyl radical footprinting data report on a region of the complex that is only peripheral to the focus on h3 and S4. Are comparable data available for the regions highlighted in the model? In the text, make an explicit connection between the footprinting and the model shown.

Response: The point of Fig. 6b was to show that S20 causes a conformational change in the helices recognized by protein S16, which is part of our model. We revised our discussion on pg. 9 to more concisely explain how hydroxyl radical footprinting results specifically support our model. To reduce confusion, we also removed panel b from Fig. 6, and added a new figure in the supplement comparing the footprinting data for this region in the presence of S20 and S17.

There are frequent discrepancies between figures and figure legends that frustrated this reader. For example, Figure 5: 5a legend ...with (green) and without (blue) S16 - there is no green. 5c med and dark blue sites cannot not be distinguished (at least in my printout and the screen). In 5c, label h3. In Figure 6, there is no 6c. Figure 1a has no black or green spheres. Figure 4a colors aren't coordinated with text.

Response: We regret the incorrect figure legends, which were not properly updated to reflect a change in our color scheme as we drafted the manuscript. The figure legends were revised to be consistent with the figures, as listed at the end of this response letter. We lightened the med blue in Fig. 5c to provide more contrast.

Reviewer #2 (Remarks to the Author):

Overall, I think the data in this paper is valuable for the construction of new experiments and for comparison to simulations. Also, I agree with the authors that there are protein-coupled changes in dynamics and that these changes are strongly-related to the hierarchy detailed in the Nomura map. But I

don't think that the data in this paper directly supports the claim of the title - "Assembly hierarchy of ribosomal complexes depends on protein-coupled RNA dynamics." The data characterizes the dynamics of the 16S in various protein bound stages, but is not able to isolate changes that are responsible for further protein binding (S16 binding in the case of this paper).

1. I think the three color FRET system developed here is well characterized and gives clean data. But, I don't think that the FRET data supports that protein-coupled changes in RNA dynamics are important for the binding hierarchy. The data related to the title is that S16 can bind more stably when S20 is already bound, which they indeed see with the FRET, but the FRET does not tell them how S20 changes the dynamics of the small subunit to allow S16 to bind. Fig 1 shows a FRET distribution +S16 vs +S16S20 which are the same. They show that S16 can (at least at higher than physiological Mg) change the dynamics of helix 3, but this is after S16 is already bound.

Response: There are three instances of hierarchy in this system: S4 enables binding of S16, S20 also enables stable S16 binding through a different allosteric path, and S16 allows S12 binding later in 30S assembly. We show that S4 changes the rRNA dynamics in a way that affects S16 recruitment. We also show that S16 binding (as well as S4) affect the dynamics of 16S helix 3 in a way that favors further assembly of the 30S ribosome, including the addition of protein S12. However, we agree that our FRET labels only directly read out the dynamics of helix 3. The effect of S20 on the rRNA is inferred from S16 recruitment, from previous footprinting data (in the discussion), and from the MD simulations that predict distinct alterations in the dynamics of the complex when S20 is bound.

We have expanded our discussion of the evidence for communication between S20 and S16 binding on pg. 8-9. We also explain on pg. 5 that the low FRET peak shifts from $E \sim 0.23$ to ~ 0.33 upon S20 binding, which is opposite the shift to $E = 0.11$ upon S17 binding (Fig. 1e-j). This shift caused by S20 may indirectly indicate the delicate changes in this region of the rRNA that are required for productive S16 binding, because when S16 is bound, the low FRET peak is similarly elevated (to ~ 0.4). That S20 and S17 have different effects on the average conformation of this region of the rRNA is directly supported by footprinting data (Fig. S7). Perhaps I should add that we repeatedly tried to label 16S helices 6 and 10, near the S20 binding site, but were unable to do so without also perturbing S20 binding. We believe this is due to the complex conformational changes that occur in this region of the rRNA during assembly.

Finally, we changed the title of the paper to: "Evolution of protein-coupled RNA dynamics during hierarchical assembly of ribosomal complexes", which we feel better captures the new findings in this paper.

2. The SHAPE analysis also seems not to target the differences in small subunit conformation allowing S16 binding because they do not compare SHAPE with S20 and without S20 in the absence of S16. They show that there are global differences with and without proteins, but do not try to isolate SHAPE differences between 16S assemblies that are favorable/unfavorable to S16 binding.

Response: Yes, the SHAPE experiments reported here focus on the effect of S16, not S20. We compare the modification pattern of the RNA when complexed with three proteins (S4, S17, S20) or with four proteins (S4, S17, S20 and S16). Because we had earlier compared the conformations of 5' domain complexes with and without S20 using hydroxyl radical footprinting, we did not repeat this analysis by SHAPE. We have revised our discussion on pg. 9 of how the hydroxyl radical footprinting data support the idea that S20 binding alters the conformation of the S16 binding site. We also inserted a figure comparing footprinting of this region in the presence of S20 and S17 (Fig. S7).

3. There is some discussion of later binding of S12 being affected by the dynamical changes imposed by S16 binding. These arguments make sense, but for this to be used in support of the central argument it must be explained what the evidence there is that S12 depends on these previous S16 binding changes. As far as I know, there are multiple events, S16 binding and the existence of the central domain before S12 can bind, which makes it difficult to identify the differences caused by S16 in isolation with S12 binding.

Response: The reviewer is correct that S12 does not stably bind the 16S 5' domain in isolation, so this means we were not able to observe S12 binding directly in our smFRET experiments. Nevertheless, it is very well established from assembly mapping experiments that S16 must bind the 16S rRNA in order for S5 and S12 to join the complex. Early footprinting experiments from the Noller lab showed that S16 affects the structure of the central domain, as well as the 5' domain. So, the two effects mentioned by the reviewer (the presence of the central domain and the conformation of h3) are linked. S12 binds over the top of the surface created by 16S h3 and h12, and at the interface between the 5' and central domains, so it is hard to imagine how S12 could bind except when h3 is in its native conformation. We revised our discussion on pg.8 to explain more specifically that S16 binding affects both h3 docking and interactions with the central domain, thus explaining why it is needed for S12 recruitment.

A couple technical questions:

4. In Fig 4 there is a typo "excitstion" in panel a. I am a bit confused by the colors of the Cy3/Cy5 excitation labels. Cy3 seems to be colored blue, but the Cy3 excitation label is written in green. Also the colors of the h4/S4/S16 don't seem to be consistent with S4 being bound to Cy5.

Response: Thank you for pointing out these errors. We have revised all Figures/Legends as listed at the end of our response.

5. Why are the shifts in h3 dynamics upon S16 binding presented as histogrammed kinetic traces? If it is the case that S16 binding changes the nature of the small subunit, then the kinetics and thermodynamic shift in h3 FRET signal should be preserved for the entire dwell time of S16 bound to the small subunit. If so, then limiting your data set to just the binding events greatly limits the available data.

Response: The FRET histograms in Figure 1e-j show the protein-induced change of the h3 conformation "at equilibrium" (over long times after protein binding). Figure 4 reports measurements in which S16-Cy3 was injected into the slide chamber during the observation. The "histogrammed kinetic traces", constructed only from the binding events in these "flow-in" measurements, compare the rates (or probabilities) of transitions to and from the high FRET state, which we can uniquely observe in this experiment. We expected to see h3 dock, perhaps stably, before S16 binds. What we observed is that h3 continues to fluctuate between its low and high-FRET states, before and after S16 binds. S16 can bind either state, but the transition probabilities change, which is what the histograms show in Fig. 4d-g. The conformational transition of h3 is not instantaneous upon S16 binding but, when averaged over molecules, the balance gradually shifts toward the native state (Fig. 4b). This indicates that S16 binding is communicated through other parts of the 5' domain RNA and induces a delayed change in h3 dynamics, as discussed in the manuscript.

Reviewer #3 (Remarks to the Author):

Pioneering work of Nomura and Nierhaus in the 1970's showed that reconstitution of bacterial ribosomal subunits from their constituent RNAs and proteins was a hierarchical process. The stable association of some proteins with rRNA depended on the prior association of others. Subsequent work by them and many others has unravelled many details of rRNA folding dynamics, allosteric effects within the rRNP, and transitions of r protein-rRNA interactions from less stable "encounter complexes" to the final stable structure, etc.. However, we still do not completely understand the details of the fundamental mechanisms by which multiple proteins assemble with a long RNA molecule. For example, what underlies the hierarchy if the proteins do not directly contact each other? Here, Sarah Woodson and her coworkers use three color single molecule FRET, molecular dynamics simulations, and SHAPE assays to uncover mechanisms of the interdependency among bacterial ribosomal proteins S4, S17, S20 and S16 and rRNA to assemble with the 5' domain of 16S rRNA.

The take-home messages are impactful to the field of ribosome assembly, and are of broader interest for those studying RNA-protein interactions. The bottom line of this work is summarized best by the authors: that binding of an r protein does not simply stabilize pre-existing native tertiary structures, but alters the

RNA dynamics to "make the search for native structure more efficient"..... "The experiments directly capture the dynamic nature of the allosteric interactions that underlie the cooperativity of ribosome assembly."

The manuscript is well written and generally accessible, even to those not aficionados of SM studies. The hypotheses, experimental rationale, and conclusions are elaborated clearly, in a stepwise fashion.

However, surprisingly, and in stark contrast, the Figure legends contain many trivial mistakes. Perhaps lost in transition to the last draft of the ms... I suggest that the authors spend some time carefully editing the legends. Some examples: Figure 1a: do the colors match those listed in the legend? Where is figure 1k? Also, since it is so critical to help readers understand this work, I suggest enlarging Fig.1a, showing the structure of the RNP domain being assembled. Fig.5: colors are not labeled correctly. Fig.6: there is no fig. 6c as indicated by the legend. Please also enlarge or highlight more the red arrow.

Please see the responses to reviewers 1 and 2. We reorganized Fig. 1 to enlarge the structure schematic in part (a) and to place part (k) next to (j) where it logically follows.

Reviewer #4 (Remarks to the Author):

In this manuscript, the authors examine the hierarchical interplay with which the ribosomal proteins S16, S4, S17, and S20 stabilize the folded 16S rRNA and assemble into the 30S ribosome, facilitating the formation of the native 30S ribosome structure. The authors utilize a combination of elegant 3-color single-molecule FRET and colocalization imaging approaches, footprinting and molecular dynamics simulations.

This manuscript represents a substantial and timely advance and reports exciting new data that fill major gaps in our understanding of those processes that enable rapid and efficient assembly of native ribosomes. Importantly, the data presented convincingly support the conclusions. In particular the single-molecule FRET experiments by the Ha laboratory are technically of high quality and well presented. This is a very challenging system to study in the context of a high-quality single-molecule experiment, and even more so since it involves 3-color FRET imaging.

However, the figures appear to be assembled in a haste and readability suffers. For example, the labeling scheme in Figure 1a is not depicted in a clear manner. Are the green spheres that are referred to in the legend of Figure 1a green or blue? Where is the sphere for Cy3 on S17 in Figure 1a? The black sphere referred to in the legend appears purple in the structure. The legend for "b-d" needs to be fixed. Similarly, the figure legend for Figure 3 states that "a-c" show fluorescence traces, while this is clearly not the case, etc.

There are a large number of such inaccuracies throughout the manuscript, and a carefully revised version would greatly help the reader to appreciate the authors' important experimental findings.

Aside from these minor issues, I believe that this will be a very important paper in helping our understanding of the hierarchical ribosome assembly move forward. As such I am happy to recommend publication of this manuscript in Nature Communications.

Response: (COMMON TO ALL REVIEWERS) We have corrected errors or minor issues in the Figures and Legends as listed below.

1. We rearranged Figure 1 to increase the size of part A and place part K more logically.
2. Figure 1 legend: The legend to the colors for Cy7 (magenta sphere); S4-Cy5 (green sphere) and Cy3 (blue spheres) was corrected.
4. Figure 1b-d: The labels were corrected.
5. Figure 3 legends: the labels on the subpanels were corrected, as was the main text.
6. Figure 4a: "Cy3 excitstion (in green)" was corrected to "Cy3 excitation (in blue)"; "Cy5 excitation (in red)" -> "Cy5 excitation (in green)"

7. Figure 5a legends: **a**, Snapshots from two molecular dynamics trajectories of h3 (cyan) and h12 (red: without protein; orange: with S4, S16, S17, S20).
8. Figure 5c: The medium blue was lightened for greater contrast with the dark blue, and the legend was edited.
9. Figure 6a: We thickened the red arrows representing the helix dynamics.
10. Figure 6 legends: parts b and c were deleted to match the corresponding change in the figure.

Reviewers' Comments:

Reviewer #1 (Remarks to the Author):

This revision is clearly written, flows logically, and has intelligible figures. Statistical analysis is appropriate.

Insights into rRNA behavior upon binding of its proteins are novel, and likely to presage other combinations of protein binding in ribosome assembly. These complicated experiments illustrate the details of molecular motions, structural and dynamical accommodations, and long-range communication in this classic RNP.

A question for the authors to consider is the use of nomenclature native and non-native. Clearly the RNA can access different conformations in the RNPs, so is there really a single native form?

Reviewer #2 (Remarks to the Author):

Related to point 1: Noting the shift in the extended basin in Figure 1 between S17 and S20 binding relative to the position in S16 was a helpful addition.

Related to point 2: Ok

Related to point 3: Ok

Related to point 4: Ok

Related to point 5: I think I understand now how Fig 4 is meant to be understood. It is a very elegant and difficult experiment to look at the details of the shift seen in Fig 1f to Fig 1j. I do though still find it difficult to see why this experiment was done, i.e. what is unexpected and could have turned out differently given Fig 1? If h3 is undocked when S16 is bound in Fig 1, there is no chance that it will be completely docked during S16 binding, S16 binding will just shift the equilibrium. And the fact that it is "delayed" is not surprising, the rate of response is limited by the barrier to h3 motion. Direct measurement of the small changes in the rates of h3 motion is a possible motivation of this experiment, but the significance of quantifying them is not addressed in the paper.

5a. In the added second to last paragraph in the Discussion the authors seem to contrast their results with allostery in proteins with the word "Nevertheless". I fail to see the conceptual

difference between a folded RNA binding a protein and shifting h3 to a more stably bound equilibrium and a protein binding a ligand and moving a distal helix to a more stably bound equilibrium.

5b. Also in that paragraph, the authors describe the proteins binding as moving the RNA through “a series barrier crossing events.” In the paper there is a single process monitored by the smFRET that can be called a barrier crossing process, the docking/undocking of h3, and as the authors show, this barrier can be crossed in both directions regardless of the proteins that are bound. It seems that the proteins shape a gradual population shift by changing the relative depth of the minima since the barrier is relatively unchanged and open/close can always be sampled. I really think this “barrier crossing” should be rethought, especially if the authors are basing their statement on this paper’s data.

Reviewer #3 (Remarks to the Author):

The authors have very carefully addressed all of my concerns.

Reviewer #4 (Remarks to the Author):

All concerns have been satisfactorily addressed and I am happy to recommend this important manuscript for publication in Nature Communications.

Responses to referee comments:

Reviewer #1 (Remarks to the Author):

This revision is clearly written, flows logically, and has intelligible figures. Statistical analysis is appropriate.

Insights into rRNA behavior upon binding of its proteins are novel, and likely to presage other combinations of protein binding in ribosome assembly. These complicated experiments illustrate the details of molecular motions, structural and dynamical accommodations, and long-range communication in this classic RNP.

A question for the authors to consider is the use of nomenclature native and non-native. Clearly the RNA can access different conformations in the RNPs, so is there really a single native form?

Response: This is a good point – the complexes are dynamic and the “non-native” flipped intermediate appears to lie on the typical path to RNP assembly, so in this sense, it is not aberrant. When we use the word “native”, what we really mean is the conformation seen in crystal structures of mature ribosomes. (Of course, mature ribosomes adopt different structures even in crystals.) For simplicity, we prefer to keep the wording as it is, but we briefly explain our use of the terms “non-native” and “native” in this context on pg. 3, with the sentence:

Productive complexes access both conformations, and in this context, we use the term ‘native’ simply to designate the conformation in the mature ribosome.

Reviewer #2 (Remarks to the Author):

Related to point 1: Noting the shift in the extended basin in Figure 1 between S17 and S20 binding relative to the position in S16 was a helpful addition.

Related to point 2: Ok

Related to point 3: Ok

Related to point 4: Ok

Related to point 5: I think I understand now how Fig 4 is meant to be understood. It is a very elegant and difficult experiment to look at the details of the shift seen in Fig 1f to Fig 1j. I do though still find it difficult to see why this experiment was done, i.e. what is unexpected and could have turned out differently given Fig 1? If h3 is undocked when S16 is bound in Fig 1, there is no chance that it will be completely docked during S16 binding, S16 binding will just shift the equilibrium. And the fact that it is “delayed” is not surprising, the rate of response is limited by the barrier to h3 motion. Direct measurement of the small changes in the rates of h3 motion is a possible motivation of this experiment, but the significance of quantifying them is not addressed in the paper.

Response: When we first planned this experiment, we initially thought that S16 would preferentially bind the high FRET state. We also considered the possibility that S16 might initially bind either the low or high FRET state, but still prevent fluctuations from the high FRET state back to the low FRET state. Of course, what we actually observed turned out to be more subtle and interesting than our initial guesses. To provide more context for this experiment, as suggested by the referee, we added a few sentences on pg. 6:

We theorized that S16 might preferentially bind the complex when it is already in the high FRET state. Alternatively, S16 might bind either the low or high FRET state, but stabilize the high FRET only after the initial encounter.

In the next paragraph, we note that S16 can (productively) bind either form of the complex – which was not a foregone conclusion at the start of this project.

5a. In the added second to last paragraph in the Discussion the authors seem to contrast their results with allostery in proteins with the word “Nevertheless”. I fail to see the conceptual difference between a folded RNA binding a protein and shifting h3 to a more stably bound equilibrium and a protein binding a ligand and moving a distal helix to a more stably bound equilibrium.

Response: We agree that there is conceptually no difference between allostery in proteins and in nucleic acids – we meant to distinguish classical models of allostery in which the change is modeled as an “all or none” shift to a new equilibrium, from newer models of allostery in which this change can be accomplished by shifts in the dynamics of the system. To avoid misunderstanding, the word “protein” was taken out of the first phrase – although the models were developed for proteins, they indeed apply to any type of macromolecule.

5b. Also in that paragraph, the authors describe the proteins binding as moving the RNA through “a series barrier crossing events.” In the paper there is a single process monitored by the smFRET that can be called a barrier crossing process, the docking/undocking of h3, and as the authors show, this barrier can be crossed in both directions regardless of the proteins that are bound. It seems that the proteins shape a gradual population shift by changing the relative depth of the minima since the barrier is relatively unchanged and open/close can always be sampled. I really think this “barrier crossing” should be rethought, especially if the authors are basing their statement on this paper’s data.

Response: Thank you for this thoughtful comment. The strongest evidence for “barrier crossing” comes from the change in the RNA dynamics when protein S4 first binds (Kim et al 2015). Here, both the speed and direction of the RNA motions change after the protein binds, and some of these changes take a long time (~0.2 s, on average), before the system settles into a slower equilibrium between the two conformations of helix 3. S10 and S17 effect a qualitative change in the complex that seem to switch into a new conformational equilibrium. The effect of S16 binding, while less dramatic, is also stretched out over 5 s in the population, suggesting a barrier of some sort. We clarified this in the text by noting that the changes happen over several seconds, and by revising the sentence to read:

This progressive change in the RNA dynamics after the initial binding event, which is even more pronounced for S4¹², suggests that the system experiences a series of barrier crossing events

Reviewer #3 (Remarks to the Author):

The authors have very carefully addressed all of my concerns.

Reviewer #4 (Remarks to the Author):

All concerns have been satisfactorily addressed and I am happy to recommend this important manuscript for publication in Nature Communications.

We thank all of the reviewers for their thoughtful and constructive comments, and we are very gratified that they find the work important.